# Rapid diversification associated with a macroevolutionary pulse of developmental plasticity

Vladislav Susoy[1], Erik J Ragsdale[1,2]*, Natsumi Kanzaki[3], Ralf J Sommer[1]*

[1]Department for Evolutionary Biology, Max Planck Institute for Developmental Biology, Tübingen, Germany; [2]Department of Biology, Indiana University, Bloomington, United States; [3]Forest Pathology Laboratory, Forestry and Forest Products Research Institute, Tsukuba, Japan

**Abstract** Developmental plasticity has been proposed to facilitate phenotypic diversification in plants and animals, but the macroevolutionary potential of plastic traits remains to be objectively tested. We studied the evolution of feeding structures in a group of 90 nematodes, including *Caenorhabditis elegans*, some species of which have evolved a mouthpart polyphenism, moveable teeth, and predatory feeding. Comparative analyses of shape and form, using geometric morphometrics, and of structural complexity revealed a rapid process of diversification associated with developmental plasticity. First, dimorphism was associated with a sharp increase in complexity and elevated evolutionary rates, represented by a radiation of feeding-forms with structural novelties. Second, the subsequent assimilation of a single phenotype coincided with a decrease in mouthpart complexity but an even stronger increase in evolutionary rates. Our results suggest that a macroevolutionary 'pulse' of plasticity promotes novelties and, even after the secondary fixation of phenotypes, permits sustained rapid exploration of morphospace.

*For correspondence: ragsdale@indiana.edu (EJR); ralf.sommer@tuebingen.mpg.de (RJS)

**Competing interests:** The authors declare that no competing interests exist.

## Introduction

Developmental (phenotypic) plasticity has been proposed to affect evolution by facilitating adaptive change (*Pigliucci, 2001*; *Schlichting, 2003*; *West-Eberhard, 2003*; *Moczek et al., 2011*) but the relevant processes resulting in evolutionary diversity remain elusive. Identification of a switch gene for a dimorphism recently confirmed the link between developmental switches and microevolutionary divergence (*Ragsdale et al., 2013*), although insights from genetic mechanisms have yet to be put into a macroevolutionary context. For example, whether plasticity accelerates evolution by allowing faster evolutionary responses (*Baldwin, 1896*; *Waddington, 1953*; *Suzuki and Nijhout, 2006*) or hinders it by allowing adaptation without the need for genetic assimilation (*Williams, 1966*) is still a matter of debate (e.g., *de Jong, 2005*; *Wund, 2012*). To know the macroevolutionary potential of developmental plasticity, objectively measured plastic traits must be compared by deep taxon sampling in a robust phylogenetic framework. Here, we test the role of developmental plasticity in evolutionary tempo and novelty by measuring change in feeding structures in a group of 90 nematodes, including *Caenorhabditis elegans*, of which some species show a mouthpart polyphenism, moveable teeth, and predatory feeding. As a result we identified both the gain and loss of a developmental dimorphism to be associated with rapid evolutionary diversification. We made the surprising finding that whereas the appearance of polyphenism coincided with increased complexity and evolutionary rates, these rates were even higher after the assimilation of a single phenotype.

The evolutionary and ecological success of nematodes is reflected by the extensive adaptation of their feeding structures, including hooks and stylets in animal- and plant-parasitic nematodes and

**eLife digest** Every animal and plant grows to a body plan that is defined by its genes. However, the body plan must be flexible enough to allow the organism to respond to whatever the world throws at it. This flexibility—known as developmental plasticity—allows an organism to change certain characteristics in order to survive in varying environmental conditions. For example, nerve cells in the brain need to be able to remodel to form memories.

It has been suggested that developmental plasticity can affect evolution because the ability to grow in different ways opens a diverse treasure trove of options from which to generate new forms and ways to exploit the environment. However, this potential had not previously been tested.

Susoy et al. looked at 90 species of roundworm that look different from one another, particularly in their mouths. Some of the worms have moveable teeth while others are simple and streamlined. Furthermore, of those examined, 23 species were found to be 'dimorphic' and have the ability to develop one of two types of mouth: either narrow or wide, depending on their prey.

Susoy et al. looked how similar the sequences of 14 genes were across all 90 species and used this information to build a family tree of how the roundworms are related to one another. Tracking which animals have dimorphic mouths on this tree produced an intriguing result: the strategy arose once in a single ancestor of the worms. Although this ability has been lost at least 10 times in the species that retained teeth, it has persisted in others through long periods of evolutionary time.

Next, Susoy et al. estimated the speed of evolution in these worms based on how quickly the characteristics of the worms' mouths had changed over evolutionary time. The gain of a dimorphic trait was associated with an increased rate of evolution and the appearance of many new species with diverse and more complex mouthparts. However, evolution was even faster where a dimorphism had been lost, even though the mouthparts generally became less complex.

Together, Susoy et al.'s findings demonstrate how developmental plasticity can introduce genetic diversity that can promote the evolution of new forms and species. The next challenges will be to find out how this genetic diversity is stored and released in the worms and to provide examples of the impact of environmental changes on developmental plasticity and shape.

teeth in predatory species. The latter adaptation is found in the genetic model *Pristionchus pacificus* and other nematodes of the family Diplogastridae, in which cuticularized teeth and predation are sometimes associated with a dimorphism (*Fürst von Lieven and Sudhaus, 2000*). Dimorphic species execute either a 'narrow-mouthed' (stenostomatous, St) or 'wide-mouthed' (eurystomatous, Eu) morph, which differ in the size, shape, and complexity of their mouthparts (*Figure 1*). In *P. pacificus*, the St and Eu morphs are advantageous for feeding on bacteria and nematode prey, respectively (*Serobyan et al., 2013*, *2014*). The dimorphism results from an irreversible decision during development, enabling a rapid optimization of morphology to the environment (*Bento et al., 2010*). This response is mediated by small-molecule pheromones (e.g., dasc#1, ascr#1) (*Bose et al., 2012*), endocrine signaling (dafachronic acid-DAF-12) (*Bento et al., 2010*), and a switch mechanism executed by the sulfatase EUD-1 (*Ragsdale et al., 2013*).

## Results

To study the tempo and mode of evolution in nematode mouthparts, we analyzed 54 species of Diplogastridae, 23 of which we found to be dimorphic. The remaining 31 diplogastrid species were identified as monomorphic. We also analyzed 33 species of other Rhabditina (*De Ley and Blaxter, 2002*), which include *C. elegans* and the closest known outgroups of Diplogastridae (*Kiontke et al., 2007*; *van Megen et al., 2009*). In contrast to Diplogastridae, all non-diplogastrid Rhabditina were monomorphic.

To test whether the dimorphism where present was a polyphenism, and not the result of genetic polymorphism (*Schwander and Leimar, 2011*), we exposed dimorphic species to cues potentially regulating their dimorphism. For assays we selected systematically inbred or genetically bottlenecked phylogenetic representatives. When exposed to signals of starvation, crowding, or the presence of nematode (*C. elegans*) prey, all species tested produced a higher number of Eu individuals in response ($p < 10^{-6}$, Fisher's exact test, for all induction experiments; *Table 1*, *Table 1—source data 1*). Thus, alternative conspecific morphs are the result of polyphenism across taxa of Diplogastridae.

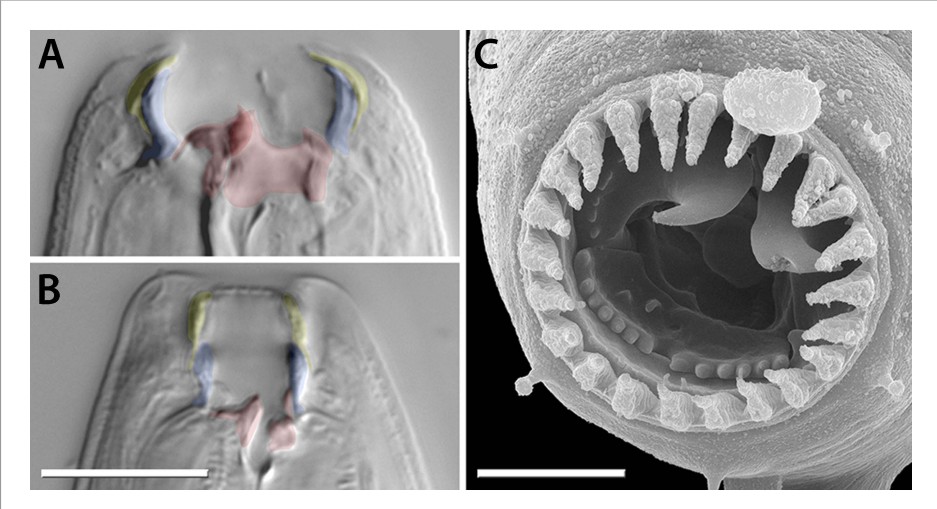

**Figure 1**. Mouth dimorphism and novelty in Diplogastridae. (**A**) The diplogastrid eurystomatous (Eu) morph, as shown here for *Parapristionchus giblindavisi*, is marked by a wider mouth, larger teeth, and often greater stomatal complexity than the stenostomatous (St) morph. (**B**) *P. giblindavisi*, St morph. False coloring in (**A** and **B**) indicates individual cuticular compartments of the mouth, providing a basis for tracking changes in homologous structures (yellow, cheilostom; blue, gymnostom; red, stegostom except telostegostom). View in (**A** and **B**) is right lateral and at same scale. Scale bar, 10 μm. (**C**) Opposing teeth, shown here for *Fictor* sp. 1, are a structural novelty of Diplogastridae and used for predatory feeding. Visible serrated plates are among other feeding innovations of Diplogastridae. Dorsal is right; scale bar, 5 μm.

To determine the order and directionality of changes in mouthpart evolution, we inferred the phylogeny of Diplogastridae and outgroups using 14 genes in an alignment of 11,923 total and 6354 parsimony-informative sites (*Figure 2A*). Because our analysis included many taxa previously not analyzed by any molecular characters, newly inferred and highly supported relationships among taxa

**Table 1**. Environmental regulation of the mouth dimorphism across diplogastridae

| Dimorphic nematode species | Treatment type | % Eu, treatment | % Eu, control | Odds ratio |
|---|---|---|---|---|
| *Allodiplogaster* sp. 1 | Prey | 100 | 0 | |
| *Allodiplogaster sudhausi* | Prey | 97 | 1 | 1080.976 |
| *Diplogasteriana* n. sp. | Starved | 24 | 0 | |
| *Fictor stercorarius* | Prey | 96 | 0 | |
| *Koerneria luziae* | Starved | 5 | 0 | |
| *Micoletzkya inedia* | Prey | 95 | 0 | |
| *Micoletzkya japonica* | Prey | 92 | 0 | |
| *Mononchoides* sp. 1 | Prey | 98 | 10 | 120.272 |
| *Mononchoides* sp. 3 | Prey | 100 | 6 | |
| *Neodiplogaster* sp. | Prey | 100 | 0 | |
| *Parapristionchus giblindavisi* | Starved | 34 | 6 | 8.428 |

The presence of prey nematode (*C. elegans*) larvae and the absence of bacterial food ('prey' treatment) induced development of the Eu morph in strains normally St-biased on an abundance of bacterial food (control). For species that could not reach adulthood on this regimen, conditions of overpopulation and starvation ('starved' treatment) similarly promoted the Eu morph. Effect size is given as the odds ratio (Fisher's exact test) where not infinite.

**Source data 1**. Environmental induction of the Eu morph in dimorphic species. Results for individual replicates (plates) are shown.

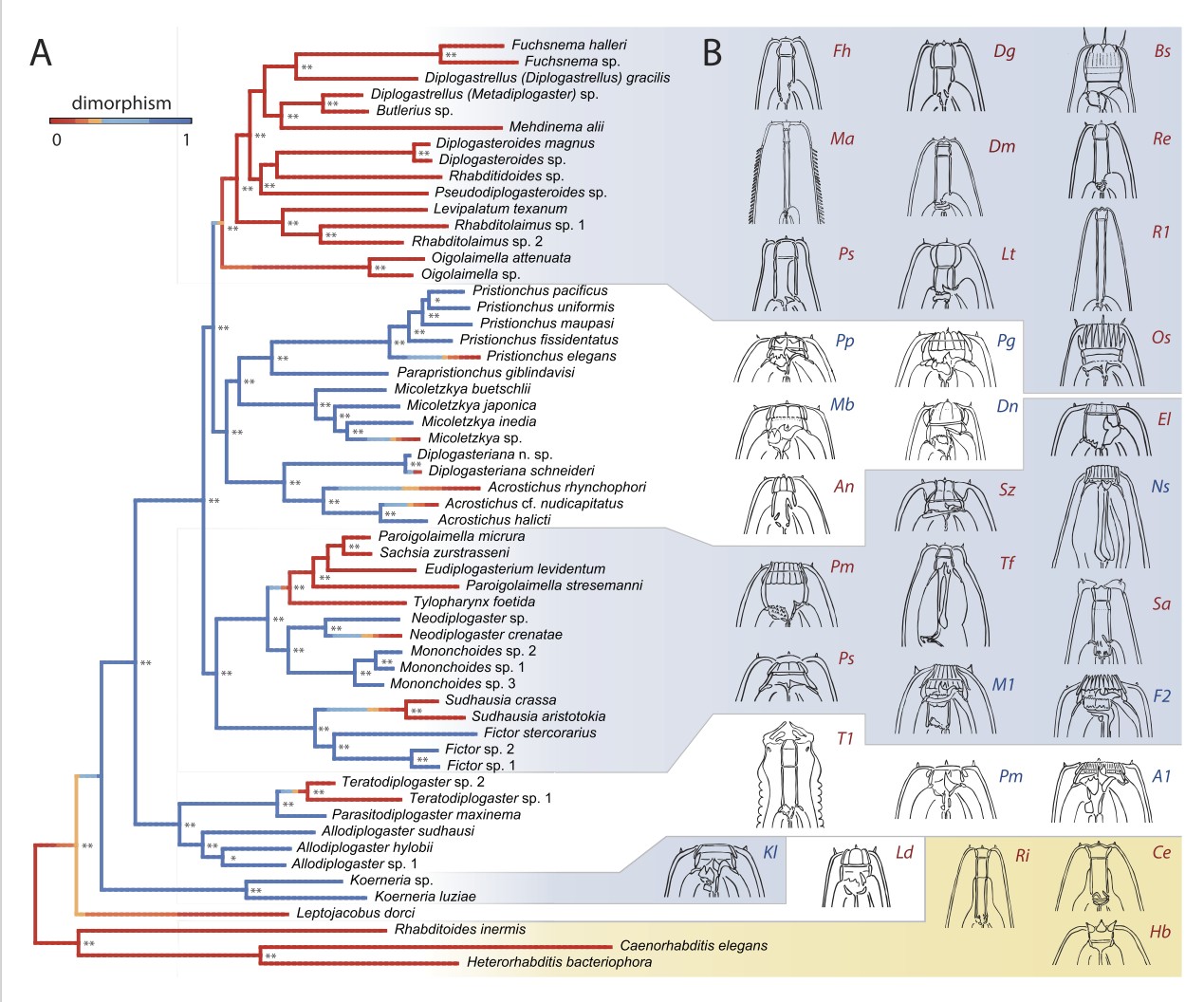

**Figure 2**. A radiation of feeding structures in diplogastrid nematodes. (**A**) Phylogenetic relationships inferred for nematodes of Rhabditina, including 54 species of Diplogastridae (*Figure 2—source data 1A,B*) from an alignment including SSU rRNA, LSU rRNA, and 11 ribosomal protein genes (for Diplogastridae, 468 kb excluding missing data), and RNA polymerase II. History of dimorphism inferred by stochastic character mapping on the set of sampled Bayesian posterior trees (consensus tree is shown). \*\*100% posterior probability (PP); \*99% PP. (**B**) Morphological diversity of mouthparts in Diplogastridae (light blue and white blocks), which are strikingly complex with respect to outgroups (yellow block). The origin of plasticity coincided with a radiation of complex feeding-forms, which variously include opposing teeth, bilateral asymmetry, and additional armature and articulations. In shape, form, and complexity, the mouths of outgroups (*Ri, Ce, Hb*) are more similar to the St than the Eu morph of dimorphic species. For dimorphic taxa, Eu morph is shown. Two-letter designations abbreviate Linnaean binomials of depicted species.

The following source data is available for figure 2:

**Source data 1**. Nematode taxa used in this study, with isolation details given.

allowed robust inferences of ancestral states. The inferred history of the mouth dimorphism revealed that it evolved once but was lost at least 10 times, and possibly 11 given the ambiguous position of *Leptojacobus dorci* (*Figure 2A*). Thus, the morphological diversity of diplogastrid mouthparts (*Figure 2B*) represents a radiation that accompanied the origin of polyphenism in those structures and involved many independent transitions to a monomorphic phenotype.

Next, we wanted to know whether the radiation of mouthparts in Diplogastridae that had dimorphism in their history represented a measurable increase in morphological variance with respect to outgroups. We quantified mouth morphology by recording 11 geometric landmarks

of the stoma that were considered homologous, as informed by fine-structural anatomy, across Diplogastridae and outgroups (*Baldwin et al., 1997*; *Ragsdale and Baldwin, 2010*) (*Figures 1A,B, 3A*). Analysis of landmark coordinates in Procrustes space for shape and form, the latter including shape + log-transformed centroid size (*Dryden and Mardia, 1998*; *Mitteroecker et al., 2004*), showed that non-diplogastrid Rhabditina occupy only a subset of the total morphospace colonized by Diplogastridae (*Figure 3A*, *Figure 3—figure supplement 1*, *Figure 3—source data 1A–D*). This represented greater disparity for Diplogastridae than for non-diplogastrid Rhabditina, whether disparity was measured as the sum of variances ($p < 10^{-5}$ when either St or both morphs represented dimorphic taxa) or by principal component analysis (PCA) volume (*Ciampaglio et al., 2001*) (*Figure 3B*, *Figure 3—source data 1E*). However, the disparity for either morph of dimorphic taxa was not different from that of non-diplogastrid Rhabditina. In contrast, diplogastrids that were secondarily monomorphic showed higher disparity than either morph in dimorphic taxa ($p < 0.02$ for both) (*Figure 3B*, *Figure 3—source data 1E*). Taken together, these findings show clear disparity differences between non-diplogastrid Rhabditina, dimorphic Diplogastridae, and secondarily monomorphic Diplogastridae.

We next tested if the observed morphospace occupation differences within Diplogastridae and across Rhabditina reflected shifts in evolutionary tempo, specifically with the gain or loss of the mouth polyphenism. Using the inferred phylogenies we measured the rate of change in shape and form (PC1) as a Brownian rate parameter under one-, two-, and three-rate parameter models (*O'Meara et al., 2006*). We found that the two-rate model that approximated different rate parameters for non-diplogastrid Rhabditina and Diplogastridae was favored over the single-rate model for both form (ΔAICc = 5.34; p = 0.01, likelihood ratio test) and shape (ΔAICc = 11.71; p < 0.001), with rates in Diplogastridae being higher (*Figure 3C,D*, *Figure 3—figure supplement 2*, *Figure 3—source data 1F,G*). Furthermore, a three-rate model that assumed a different rate parameter for each of the three nematode groups had the greatest fit compared with either a single-rate model (ΔAICc = 9.18, p = 0.038 for form; ΔAICc = 14.79, p < 0.001 for shape) or a model that assigned a different rate category to dimorphic diplogastrids (ΔAICc = 9.32, p < 0.01 for form; ΔAICc = 15.27, p < 0.001 for shape), and rates in monomorphic Diplogastridae were the highest (*Figure 3C*). For form evolution in particular, a two-rate model that assumed a different rate parameter for monomorphic Diplogastridae was a better fit than all other models, including that with a single category for Diplogastridae (ΔAICc = 5.23). Congruent with these results, a comparison of posterior densities of rate estimates from the Bayesian sampling of a multirate Brownian-motion process (*Eastman et al., 2011*), which were extracted for individual nematode groups, indicated elevated rates of evolution in Diplogastridae relative to non-diplogastrid Rhabditina, with rates in secondarily monomorphic lineages being the fastest (*Figure 3—figure supplement 3*, *Figure 3—source data 1H*). Thus, our analyses of evolutionary rates show that diversification of shape and form in Diplogastridae increased with the appearance of the mouth plasticity but were highest after its subsequent loss.

We then wanted to know whether developmental plasticity also correlated with the complexity of mouthparts that distinguishes Diplogastridae from their closest relatives (*Figure 2B*). We tabulated complexity for all taxa by recording the number of stomatal structures or 'cusps', adapting a concept of complexity commonly applied to the dentition of vertebrates (*Harjunmaa et al., 2012*). Namely, we scored all structures or articulations that formed a <135° vertex with the wall of the stoma (*Figure 4—figure supplement 1*, *Figure 4—source data 1A*), summing the total to an index that was invariable for all specimens of a given species or, in dimorphic species, a particular morph (here, Eu). We then tested for phylogenetic correlations of this complexity index with the presence of plasticity. Plasticity was strongly correlated with greater complexity, as shown by their covariance tested either under the threshold model (*Felsenstein, 2012*) (r = 0.78, confidence interval 0.57–0.93) or a constant-variance random-walk model (r = 0.45; log Bayes factor = 20). Given the character histories of known taxa (*Figure 4*), this result reveals that the gain of the polyphenism was simultaneous with the onset of high complexity, including the origin of opposable teeth. In contrast, the loss of the polyphenism in monomorphic Diplogastridae was associated with a subsequent decrease in complexity.

## Discussion

Our results provide original statistical and phylogenetic support for a role of developmental plasticity in evolutionary diversification. They are also congruent with a simple model for the role of plasticity in this process. First, the appearance of bimodal plasticity coincides with a burst of complexity and

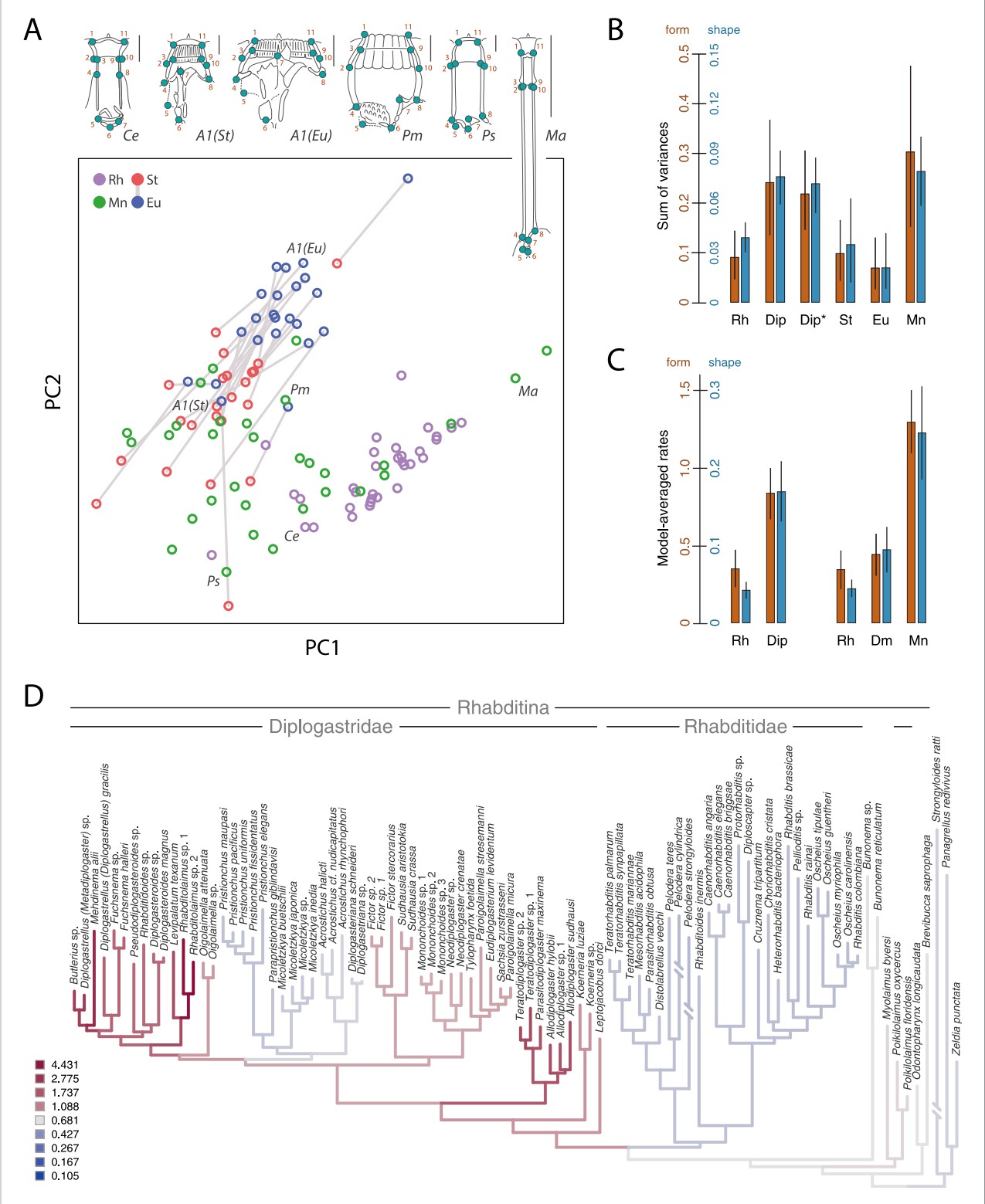

**Figure 3**. Developmental plasticity, morphological disparity, and evolutionary tempo in diplogastrid nematodes. (**A**) Stomatal morphology and positions of 11 two-dimensional landmarks (taxa coded in *Figure 2*). Below is a projection of the first two principal components of stomatal shape-space. Purple circles represent non-diplogastrid Rhabditina (Rh), green circles mark monomorphic Diplogastridae (Mn); blue and red circles connected by lines mark St and Eu morphs, respectively, of dimorphic Diplogastridae. (**B**) Phenotypic disparity of non-diplogastrid Rhabditina (Rh), Diplogastridae (Dip, dimorphic taxa are represented by St morph; Dip*, by both morphs), and individually of St, Eu, and monomorphic (Mn) Diplogastridae, as estimated by the sum of

*Figure 3. continued on next page*

*Figure 3. Continued*

variances on shape- and form-space axes. Bars show mean values from 10,000 bootstrap replicates. Whiskers represent a 95% confidence interval. (**C**) Model-averaged relative estimates of evolutionary rates, as estimated under a Brownian motion model. Both a two-rate model (left) and a three-rate model (right) are shown (Dm, dimorphic Diplogastridae as represented by St morph). Bars are mean rates calculated across 5000 reconstructions of dimorphism history and 500 trees. Whiskers represent the standard deviation. (**D**) Rate estimates of stomatal form evolution in Rhabditina. In dimorphic taxa, rates are for St morph. Branch color indicates rates of evolutionary change; posterior rates are color-coded in legend.

The following source data and figure supplements are available for figure 3:

**Source data 1**. Results from analyses of principle components, disparity, and evolutionary rates.

**Figure supplement 1**. Projections of the first two principal components of Procrustes morphospace of stomatal landmarks.

**Figure supplement 2**. Rate estimates of stomatal shape evolution in Rhabditina.

**Figure supplement 3**. Posterior densities of rates of stomatal form and shape evolution in Rhabditina.

---

increase in evolutionary tempo. By this model, developmental plasticity can facilitate novel structures and their associated developmental networks (*West-Eberhard, 2003*), as well as new complexity in behavioral or enzymatic function, thereby providing additional substrate for future selection. Following this macroevolutionary 'pulse' of plasticity, the secondary loss of plasticity is accompanied by a decrease in complexity but a strong acceleration of measured evolutionary rates, which in our study were most pronounced in form change. The surprising limitation of rates in dimorphic relative to secondarily monomorphic lineages might be explained in part by genetic correlation (*Cheverud, 1996*), or the inability of overlapping genetic programs controlling alternative phenotypes to completely dissociate. We speculate that, where correlated morphologies were initially governed by a dimorphism, assimilation of a single morph would then give the freedom for single phenotypes to specialize and diversify, a phenomenon proposed as developmental 'character release' (*West-Eberhard, 1986*).

A complementary means by which evolutionary rates increase after the loss of plasticity may be through the release of genetic variation built up as a by-product of relaxed selection (*Kawecki, 1994*; *Snell-Rood et al., 2010*; *Van Dyken and Wade, 2010*). This possibility might be realized through the following scenario. If populations experience fluctuating environments and alternative mouth morphologies confer fitness advantages in those environments, then environmental sensitivity (i.e., plasticity) will be maintained (*Moran, 1992*). The presence of plasticity necessarily leads to relaxed selection on genes underlying the production of either trait, particularly those downstream of a developmental switch, facilitating the accumulation of genetic variation (*Van Dyken and Wade, 2010*). If populations then encounter a stable, predictable environment, promoting the loss of plasticity (*Schwander and Leimar, 2011*), this variation can be selected and refined by constitutively exposing a single morph to that environment. This would allow more rapid evolution of novel phenotypes than would be possible through the generation and selection of new genetic variation (*Barrett and Schluter, 2008*; *Lande, 2009*), thereby allowing rapid shifts to alternative niches such as novel diets (*Ledón-Rettig et al., 2010*). Combined with the ability of fixed morphs to more efficiently reach their fitness optima as permitted by character release, variation accumulated during periods of plasticity would thus enable rapid phenotypic specialization and diversification. Although accelerated rates of divergence due to built-up variation and character release should ultimately decline in monomorphic lineages (*West-Eberhard, 2003*; *Lande, 2009*), the net result would be an extreme radiation of forms, as has occurred in diplogastrid nematodes.

In conclusion, the historical presence of polyphenism is strongly associated with evolutionary diversification. The degree to which the correlations observed are due to causation is presently unclear, although recent mechanistic advances in *P. pacificus* demonstrate the promise of functional genetic studies to test the causality of rapidly selected genes directly. Further work might also reveal that additional underlying causes, such as previously unseen ecological opportunities or selective pressures, may have jointly led to both complexity and plasticity. However, the simplicity of our results makes our proposed model sufficient to explain the observed correlations. We therefore hypothesize

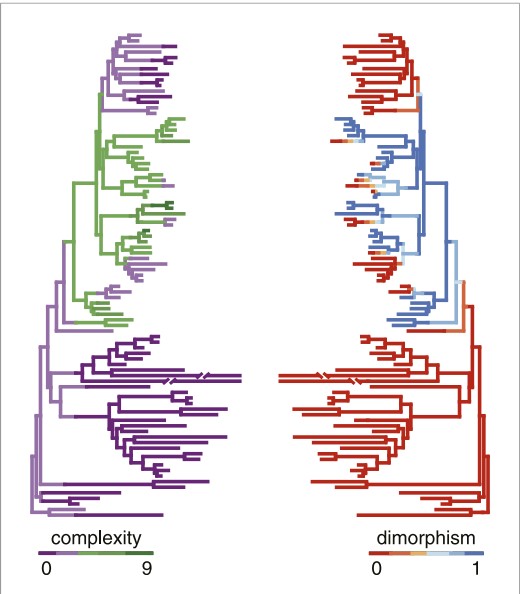

**Figure 4**. Correlation of polyphenism and complexity of nematode mouthparts. Painted branches show congruence of simulated character histories of dimorphism (right tree; 0 = absent, 1 = present) and stomatal complexity (left tree; complexity index ranges from 0 to 9). Covariance tests (see text) show that the apparent phylogenetic correlation between dimorphism and complexity is significant.

The following source data and figure supplements are available for figure 4:

**Source data 1**. Matrix of structures tabulated to measure stomatal complexity.

**Figure supplement 1**. Tabulating complexity of nematode mouthparts.

that developmental plasticity is required to cross a threshold of complexity that affords the degrees of freedom necessary for further diversification of form, and even after the assimilation of monomorphy this diversification can continue to be realized. The difference in rates between ancestrally and secondarily monomorphic lineages suggest a deciding role for a history of plasticity in diversification. It is possible that the processes inferred to accompany the gain of plasticity apply also to other systems with taxonomically widespread polyphenism, which sometimes likewise show a general coincidence of plasticity and diversity (e.g., *Emlen et al., 2005*; *Pfennig and McGee, 2010*). In principle, the model we propose can be generalized to other systems through dense taxon sampling, a resolved phylogeny, and quantification of alternative morphologies.

## Materials and methods

### Nematodes

To investigate evolutionary rates, complexity, and character histories, we densely sampled nematodes of Rhabditina *sensu De Ley and Blaxter (2002)* (= Clade V *sensu Blaxter et al., 1998*). Isolation details for all analyzed nematode taxa for which original sequence data were obtained are given in *Figure 2—source data 1A*. Our taxonomic nomenclature follows previous systems (*Andrássy, 1984*, *2005*; *Sudhaus and Fürst von Lieven, 2003*) with additional genera described since those publications (*Ragsdale et al., 2014*). Our dataset included 54 species of Diplogastridae, in addition to 33 nematode species from all closest known outgroups to the family: 'Rhabditidae' sensu *Sudhaus (2011)*, Brevibuccidae, Bunonematidae, Myolaimidae, and Odontopharyngidae. In the present study, 'non-diplogastrid Rhabditina' refers to the latter five families together. Rhabditidae were sampled such that they spanned all major clades of that group as identified in a previous study (*Kiontke et al., 2007*): the *Mesorhabditis* group and non-*Mesorhabditis* 'pleiorhabditids'; *Caenorhabditis*, the four deepest lineages of the *Rhabditis* group, and the remaining two deepest lineages of 'eurhabditids'; *Rhabditoides inermis*, a possible immediate outgroup to Diplogastridae; *Poikilolaimus*, the putative sister group to all other Rhabditidae and nested taxa. Three Clade IV (*Blaxter et al., 1998*) nematode species were included as outgroups in the dataset.

### Phylogenetics

#### Dataset assembly

The phylogeny of Diplogastridae was inferred from concatenated alignments of 18S and 28S rRNA genes and 11 ribosomal protein-coding genes of 90 taxa. Genomic DNA was extracted from individual specimens and total RNA from 15–45 individuals per species (*Figure 2—source data 1B*). Genes of interest were amplified individually, and sequencing reactions were performed as previously described (*Mayer et al., 2009*). Sequences were assembled using Geneious 6.1.4. Sequences for 18S, 28S, ribosomal protein, and RNA polymerase II genes, which were either original in this study or retrieved from public databases, were included for non-diplogastrid Rhabditina and outgroups. 18S

and 28S rRNA sequences were aligned using the E-INS-I algorithm and default settings in MAFFT 7.1 (*Katoh and Standley, 2013*). Alignments were manually refined, and poorly aligned regions were eliminated manually. Alignments of 18S and 28S rRNA genes were 1598 and 3155 bp long, respectively, and included 859 and 1616 parsimony-informative sites. Sequences of each of the 11 ribosomal protein genes were aligned individually using default settings in Muscle 3.8 (*Edgar, 2004*) and were realigned by predicted translation; alignments were manually refined and stop-codon sites removed. The concatenated alignment of 11 ribosomal protein genes was 5475 bp long and included 2970 parsimony-informative positions. Aligned sequences for Diplogastridae contained 444 kb without missing data. The final dataset of diplogastrid sequences was more than four times larger than that used in the previously most inclusive phylogenetic study of the family (*Mayer et al., 2009*), and it included over three times as many species and twice as many diplogastrid genera. In the final concatenated alignment of rRNA and ribosomal protein genes for all diplogastrid species, the proportion of missing data was 20%, with a minimum of 70% of nematode species sampled per gene. The dataset of all taxa had 667 kb excluding missing data and was 11,923 bp long (*Supplementary file 1*), in which the fraction of missing data was 38%.

## Inference methods

The phylogeny was inferred under Bayesian and maximum likelihood (ML) optimality criteria as implemented in MrBayes 3.2.2 (*Ronquist et al., 2012*) and RAxML 7.3 (*Stamatakis, 2006*), respectively. All inferences were performed on the CIPRES Science Gateway (*Miller et al., 2006*). For Bayesian inference, the dataset was partitioned into four subsets: two for 18S and 28S rRNA genes, which were analyzed using a 'mixed' $+ \Gamma$ model, and the third and fourth for the combined ribosomal protein genes and RNA polymerase II, respectively, which were analyzed under a codon $+ \Gamma$ model. Model parameters were unlinked across partitions. Four independent analyses, each containing four chains, were run for 55 million generations, with chains sampled every 1000 generations. After confirming convergence of runs and mixing of chains using Tracer 1.6 (*Drummond and Rambaut, 2007*), the first 50% generations were discarded as burn-in and the remaining topologies summarized to generate a 50% majority-rule consensus tree. For the ML analysis, our partitioning scheme divided the dataset into three subsets: two for the 18S and 28S rRNA genes, which were each analyzed using a GTR $+ \Gamma$ model, and the third subset for translated ribosomal protein and RNA polymerase II genes, analyzed under an inverse-gamma (IG) $+ \Gamma$ model. The latter model was selected based on an amino-acid substitution-model test as implemented ProtTest 3 (*Darriba et al., 2011*). 100 independent ML searches initiated with random starting trees were performed. Support values for the best-scoring tree were estimated from 1000 iterations of non-parametric bootstrapping.

## Presence of polyphenism

We identified nematode species as dimorphic or monomorphic by screening at least 200 individuals in cultured populations under both well-fed and starved conditions, the latter of which is known to induce the Eu morph in *P. pacificus* (*Bento et al., 2010*). Dimorphism was diagnosed by the presence of morphs that differed (i) in the width and aspect ratio of the stoma and (ii) in the prominence and sclerotization of mouth structures (*Fürst von Lieven and Sudhaus, 2000*; *Serobyan et al., 2013*). In all examined species with mouth plasticity, the plasticity was discrete with no observed (and hence presumably rare) intermediate forms or reaction norms for morph-diagnostic morphology. Furthermore, each of the two morphs was stereotypic for a given species, such that morphology did not qualitatively vary with different induction cues. The mouth plasticity was therefore a discrete dimorphism of constant morphs in all species with the plasticity, consistent with previous observations of *P. pacificus*, for which multiple levels of starvation, pheromones, hormones, transgenes, enzyme-inhibiting salts, or environments previously experienced by wild-caught specimens all induced either of two morphs, albeit in differing ratios (*Bento et al., 2010*; *Bose et al., 2012*; *Ragsdale et al., 2013*; *Serobyan et al., 2013*). For species that could not be brought into culture (annotated as 'nc' in *Figure 2—source data 1A*), all of which were monomorphic, observations of collected isolates were corroborated by comprehensively reviewed previous taxonomic studies (*Sudhaus and Fürst von Lieven, 2003*) to confirm the absence of dimorphism. Taken together, previous reports and our own collections demonstrated that such species were monomorphic across multiple populations and environmental conditions. In each of the five cases of recent losses, namely those inferred to have occurred on a terminal branch within Diplogastridae, the assimilated morph was identified as the St morph. However, for inferred

ancient losses of the dimorphism, derived morphology made the homology of the assimilated morph impossible to determine reliably. Therefore, our analyses identify monomorphic and dimorphic taxa without distinguishing which of the two morphs was lost or assimilated.

## Environmental induction of alternative morphs

To test whether the mouth dimorphism of diplogastrid nematodes was an environmental polyphenism and not genetic polymorphism, we exposed dimorphic species to environmental conditions potentially influencing expression of the two alternative mouth phenotypes. Specifically, we tested species (strains) with high frequency of St morph for environmental induction of the Eu morph. Although all strains tested had been kept in laboratory culture for at least one year prior to experiments, several strains (*Allodiplogaster sudhausi*, both *Micoletzkya* spp., *Parapristionchus giblindavisi*) were additionally inbred systematically for 10 generations.

In our first assay (*Table 1—source data 1*), 7 fertile St females or hermaphrodites (5 for *Allodiplogaster sudhausi*) were transferred from a stock culture well-fed with bacteria onto an NGM plate (no peptone, no cholesterol) supplied with approximately 70,000–100,000 arrested *C. elegans* larvae. In parallel, the same number of St females or hermaphrodites was transferred onto NGM plates with the same species of bacteria as that on stock culture plates: this was OP50 for most species, although some species (i.e., *Micoletzkya* spp.) required different bacterial strains to reproduce. Nematodes were allowed to feed on the provided food, lay eggs, and develop in the following generation. The mouth phenotype of all F1 females or hermaphrodites was scored when those individuals reached adulthood (5–10 days, depending on the species). Experiments were performed in triplicate for each species.

Because some species could not develop in the absence of microbial food, we employed a second strategy to test for environmental induction of the Eu morph in those strains. In this assay (*Table 1—source data 1*), 10–15 fertile females were transferred to plates seeded with a 500 µl bacterial lawn. After the time necessary for the populations of a species to complete one generation following the visible depletion of a bacterial lawn (*Diplogasteriana* n. sp., 6 weeks; *Koerneria luziae*, 2.5 weeks; *P. giblindavisi*, 2 weeks), adult females were screened for their mouth phenotype. In parallel, nematodes of the same species were maintained in well-fed culture, being transferred (10–15 females per replicate) to a new bacterial lawn, the next generation being screened for the mouth phenotype after 1 week. All adult females up to a sample size of 200 per plate were screened. Experiments were performed in triplicate for each species.

For both assays, significant differences in morph ratios between prey-fed and bacteria-fed nematodes were calculated using Fisher's exact test with the total number of assayed individuals pooled across replicates. Effect sizes of differences were estimated as the odds ratio by Fisher's exact test. The percentage of the Eu morph per treatment per species is reported in *Table 1* for pooled samples.

## History of dimorphism

To infer the evolutionary history of the stomatal dimorphism, we used stochastic character mapping (*Nielsen, 2002*; *Huelsenbeck et al., 2003*) as implemented in SIMMAP 1.5 (*Bollback, 2006*). This approach estimates probabilities of the states along phylogeny under continuous-time Markov models, incorporating uncertainty in tree topology, branch length, and ancestral character states. The best-fitting parameters of morphology priors, the overall substitution rate prior (gamma distribution prior), and the bias prior for two-state characters (beta distribution prior) were estimated using a Markov-chain Monte Carlo (MCMC) method as also implemented in SIMMAP. These calibration analyses were run for 500,000 generations, sampling the chain every 100 generations, using a 50% majority rule consensus tree summarized from the Markov chains of the Bayesian phylogenetic analysis; the first 50,000 generations were discarded as burn-in. For stochastic character mapping, 500 trees were randomly sampled, with the help of Mesquite 2.75 (*Maddison and Maddison, 2011*), from trees generated during the MCMC runs. The number of discrete categories, $k$, was set to 90 and 31 for the gamma and beta distributions, respectively. Trees were rescaled to a length of one before applying priors on the overall rate. For analyses of evolutionary rates and complexity correlation, 10 character histories were simulated on each of the 500 trees. The density maps of the dimorphism history (*Figures 2A, 3E*) were generated by summarizing posterior densities from 500 simulations of character histories on the ML tree in the R package phytools 0.3-72 (*Revell, 2012*).

## Geometric morphometrics

To capture stomatal morphology, 11 fixed two-dimensional landmarks were placed at locally defined boundaries or points within the stoma (*Figure 3A*). Landmarks consisted of boundaries or points that were considered homologous across Rhabditina as predicted by fine-structural anatomy (*Baldwin et al., 1997*; *Ragsdale and Baldwin, 2010*); stomatal terminology follows *De Ley et al. (1995)*. Type-1 landmarks were recorded at the ventral and dorsal boundaries of the cheilostom with labial tissue (landmarks 1 and 11, respectively), the ventral and dorsal boundaries between the cheilostom and gymnostom (2 and 10, respectively), the ventral and dorsal boundaries between the gymnostom and stegostom (4 and 8, respectively), the posterior boundary of the dorsal telostegostom (6), and the dorsal gland orifice (7); type-2 landmarks included the anterior apex of the ventral and dorsal gymnostom (3 and 9, respectively) and the apex of medial curvature of the subventral telostegostom (5). To exclude contribution of the third dimension to morphometrics, all landmarks were recorded in exactly lateral view, as guaranteed by the body habitus of slide-mounted nematodes, that is, their sinusoidal spread along the sagittal plane.

For 68 nematode species, landmarks were recorded for multiple live specimens, which were mounted on 5% agar pads with 8 µl of 0.25 M sodium azide added as an anesthetic. Microscopy was performed using a Zeiss Axio Imager.Z1 equipped with a Spot RT-SE digital camera. Landmark positions were marked using live-view mode in Metamorph 7.1.3 (Molecular Devices, Sunnyvale, CA, USA), and after image acquisition they were digitalized using tpsDig2 (*Rohlf, 2008*). For 22 species, we used video vouchers and images from published sources for digitalization of landmarks. Our complete morphometric dataset consisted of 522 images and 90 nematode species (an average of 4.8 images per species or morph). Landmark positions and centroid sizes (square root of the sum of squared distances of landmarks to their centroid) were averaged for each species (or each morph for dimorphic species), whereafter landmarks were Procrustes-superimposed using MorphoJ (*Klingenberg, 2011*).

We used two approaches to analyze landmarks. First, we simultaneously accounted for variation in both stomatal shape and size by performing Procrustes form-space (size-shape space) analyses (*Dryden and Mardia, 1998*; *Mitteroecker et al., 2004*). In this approach, Procrustes shape coordinates, which are the result of landmark centering, rotation, and scaling, are augmented by the natural-logarithm-transformed centroid size (i.e., as calculated prior to scaling) and subjected to principal component analysis (PCA). PCA on the Procrustes shape coordinates matrix was performed with an additional column appended containing log-transformed centroid size data using the 'prcomp' function in R 3.0.2 package Stats (*R Development Core Team, 2013*). In the second approach, we performed PCA analysis on Procrustes shape coordinates to reconstruct Procrustes shape-space (*Rohlf and Slice, 1990*). In contrast to form-space, shape-space in principle minimizes the effects of allometry, offering an alternative way to measure morphological change. When data for all species and morphs were combined (*Figure 3A*), the first and the second PC axes of form-space accounted for approximately 73% and 16% (68% and 12% for shape-space), respectively, of the variance. Thus, the cumulative proportion of the overall variance explained by PC1 and PC2 axes was 88% and 81% for form- and shape-space, respectively (*Figure 3—source data 1A,B*). In form-space analyses, loadings of the log centroid size onto PC1 and PC2 axes were 0.91 and 0.41 (*Figure 3—source data 1A,B*).

In addition to the PCA above, we performed phylogenetic PCA on both form and shape matrices for evolutionary rate analyses (*Revell, 2009*) to account for phylogenetic non-independence of morphometric data. The St morph represented dimorphic species in this PCA (*Figure 3—source data 1C,D*). Disparity analyses included several components of the standard PCA were retained (see below). All other analyses, which comprised phylogenetically corrected inference and tests of evolutionary rates requiring individual variables, used scores along the first PC axis of each phylogenetic PCA and which explained the vast majority of variance in either form or shape.

## Disparity

Morphological variation (disparity) was examined in three groups, namely non-diplogastrid Rhabditina, dimorphic Diplogastridae, and monomorphic Diplogastridae. We used two approaches to investigate disparity: (i) the sum of univariate variances on form-space axes (multivariate variance) and (ii) PCA volume (*Figure 3—source data 1E*). These methods capture different aspects of morphological diversity and both are based on morphological distance measures, although neither controls for phylogenetic non-independence. The sum of variances, a variance-based metric, provides

an estimate of degree of difference among species in Procrustes morphospace. Alternatively, PCA volume gives an estimate of the amount of morphospace occupied by species; it is calculated as the product of the eigenvalues of the cross-distance matrix, divided by the square of the number of species. The sum of variances was previously shown by simulation-based studies to be relatively insensitive to variation in sample size, and both methods have relatively low sensitivity to missing data (*Ciampaglio et al., 2001*). The analyses were performed using the MATLAB package MDA (*Navarro, 2003*). PC axes that explained more than 5% of the overall variance (2 for form, 3 for shape) (*Figure 3—source data 1A,B*) were retained for calculations of the sum of variances and PCA volume. Rarefaction was performed to correct for sample-size dependence (*Ciampaglio et al., 2001*), such that the sample size was standardized to the number of species in the smallest group compared. To calculate means of disparity estimates, their standard deviations, and their 95% confidence intervals, 10,000 bootstrap replicates were performed. For pairwise comparisons of the sum of variances between groups, two-tailed p-values were estimated using 100,000 bootstrap replicates.

## Evolutionary rates

We used two comparative methods that employ a Brownian motion (BM) model to estimate and compare rates of evolution of stomatal morphology among different nematode lineages: (i) a ML-based non-censored rate test (*O'Meara et al., 2006*) and (ii) a Bayesian reversible-jump approach (*Eastman et al., 2011*). In these approaches, the rate of evolution is measured as a rate parameter for the BM process by weighting the magnitude of change of the trait per unit of 'operational time' (*Pagel, 1997*). In our analyses, operational time was set to inferred genetic distance, that is, branch lengths inferred in our Bayesian phylogenetic analysis of four partitions of the 14 included genes. This metric is supported as an appropriate measure of time by mutation accumulation line experiments, which have indicated rates of molecular evolution to be nearly identical between distantly related nematodes of Rhabditina (*Weller et al., 2014*). Absolute time was not used because (i) relevant fossil data are not available to calibrate dates in the phylogeny and (ii) the number of generations per year is assumed to differ dramatically between nematode species due to differences in generation time and, given ecological differences (*Herrmann et al., 2006*; *Kiontke et al., 2011*), presumed lengths of diapause (dauer) stages.

### Non-censored rate test

To investigate how rates of morphological (form and shape) evolution change in the presence of plasticity, we estimated the relative fit of one-, two-, and three-rate parameter models using the 'Brownie.lite' function in the R package phytools 0.3-72 (*Revell, 2012*) (*Figure 3—source data 1F*). Five BM models were tested: (i) a single rate model that approximated the same rate parameter for non-diplogastrid Rhabditina, dimorphic Diplogastridae, and monomorphic Diplogastridae (1,1,1 model); (ii) a two-rate parameter model that assigned one rate category to non-diplogastrid Rhabditina and a different category to dimorphic and monomorphic Diplogastridae together (1,2,2 model); (iii) a two-rate model that approximated one rate parameter for non-diplogastrid Rhabditina and monomorphic diplogastrids but a different rate parameter for dimorphic Diplogastridae (1,2,1 model); (iv) a two-rate model that assumed the same rates for non-diplogastrid Rhabditina and dimorphic Diplogastridae but different rates for monomorphic Diplogastridae (1,1,2 model); (v) a three-rate model that assumed different rate parameters for each of the three nematode groups (1,2,3 model). We assessed the relative fit of models by comparing second-order Akaike Information Criterion (AICc) values (*Figure 3—source data 1F*). If the difference in values (ΔAICc) was greater than 4, the worse-fitting model was considered much less supported (*Burnham and Anderson, 2002*). Additionally, nested models were compared using a hypothesis-testing likelihood-ratio approach, that is, using a chi-square distribution (*Figure 3—source data 1F*; p-values are also given in main text). Tests were performed on 5000 trees with mapped character histories, which were randomly sampled from posterior distributions of post-burn-in trees generated by the MCMC runs of the phylogenetic analysis.

### Bayesian sampling of shifts in trait evolution

We investigated variation in evolutionary rates across lineages of Rhabditina using a Bayesian reversible-jump approach (*Eastman et al., 2011*) as implemented in the R package Geiger 1.99-3 (*Harmon et al., 2008*) (*Figure 3—source data 1H*). This method estimates posterior rates of continuous trait evolution along individual branches of the phylogeny using reversible-jump MCMC sampling of a multirate BM process, without the need for specifying hypotheses a priori about the location of rate shifts. To achieve mixing of MCMC chains, we calibrated the proposal width using the

function 'calibrate.rjmcmc' and running the chain for 1 million generations, after which we used Tracer 1.6 to confirm mixing. Three MCMC analyses were then performed, with 30 million generations each, using the function 'rjmcmc.bm'. Analyses were run under a relaxed-BM model with the number of local clocks constrained to three and the proposal width set to 1.5. Chains were sampled every 5000 generations, the first 25% of generations was discarded as burn-in, and Tracer 1.6 was used to confirm chains mixing and convergence. Results from the three independent runs were combined, and weighted posterior rates of individual branches within each of the compared categories were extracted. The highest posterior density (HPD) intervals and means were estimated for the three nematode groups (*Figure 3—figure supplement 3*) and were compared using a two-tailed randomization test to determine whether posterior rates were different among groups (*Figure 3—source data 1H*).

## Dimorphism and stomatal complexity

### Tabulating complexity

To establish an index for the complexity of nematode mouthparts, we scored the total number of observed cuticular 'cusps' (*Harjunmaa et al., 2012*) and articulations, that is, structures projecting independently within the stoma. We define a 'structure' herein as any geometric deviation that is marked by a physical vertex of <135° from the cylindrical walls of the stoma or from the arched anterior margins of the pharyngeal radii (*Figure 4—figure supplement 1*). All recorded structures were discrete and stereotypic, that is, always present or absent, for each species or morph (for dimorphic taxa, the Eu morph was analyzed). Because such structures take a variety of shapes, all recorded structures are for clarity presented as a presence/absence character matrix (*Figure 4—source data 1*), which includes structures consistent with previous reports (*Fürst von Lieven, 2000*; *Fürst von Lieven and Sudhaus, 2000*; *Sudhaus and Fürst von Lieven, 2003*; *Kanzaki et al., 2012*; *Herrmann et al., 2013*; *Ragsdale et al., 2014*). Iterative structures (i.e., serratae, rods, points, warts, serial denticles, and divisions of stomatal wall) were conservatively scored as a single structure, because such iterative structures were always co-dependent and were sometimes (i.e., for denticles, serratae, and warts) variable in number among individuals of a single species. Furthermore, such structures show that this additional within-character complexity correlates with size, analogous to what is observed in mammalian tooth development (*Harjunmaa et al., 2014*) or what might otherwise be expected in area-dependent patterning (*Turing, 1952*). Therefore, to minimize the effects of size on complexity in our analyses, all tabulated structures were those that were unique and constant within a species and which could be assigned homology where present in multiple species. Additionally, structures that bore multiple vertices or 'secondary complexity' distal to its deviation from the stoma (i.e., teeth, which could have multiple bends or peaks) were also coded as single structures. Finally, any character present as identical, symmetrical duplicates, which was due to the presence of two subventral sectors and hence also developmental co-dependence, was scored as a single structure. Examples of stomata with all of their structures recorded and labeled are shown in *Figure 4—source data 1*.

All Diplogastridae and some non-diplogastrid Rhabditina were observed by differential interference contrast (DIC) microscopy. For other taxa, morphology was scored from published DIC video vouchers, DIC micrographs, and drawing interpretations; stomatal morphology for genera of Rhabditidae was additionally confirmed according to a recent key for the family (*Scholze and Sudhaus, 2011*). Observed structures comprised a total of 25 characters. For the set of analyzed taxa, the complexity index ranged from 0 to 9.

### Character correlation

We tested for a correlation between presence of mouth dimorphism and stomatal complexity using the dataset that included all 87 species of Rhabditina and the threshold model (*Felsenstein, 2012*) as implemented in the R package phytools 0.3-72 (*Revell, 2012*). We ran 50 analyses of 500,000 generations each and using trees randomly sampled from the posterior distributions of trees generated by the phylogenetic analysis in MrBayes 3.2.2. After confirming chain convergence and discarding 25% of the posterior samples as burn-in, the outputs of the analyses were combined and used to calculate the maximum likelihood estimation of the correlation coefficient. The R package coda 0.16-1 (*Plummer et al., 2012*) was used to compute the highest posterior density intervals of those estimates.

Additionally, we tested for correlation between dimorphism and stomatal complexity by Bayesian MCMC sampling as implemented in BayesTraits V2 (beta) (*Pagel and Meade, 2013*). For this test, a constant-variance random-walk model was invoked. The regression coefficient was estimated as the ratio of covariance between dimorphism presence and complexity index to the variance of dimorphism presence. Significance of the trait correlation was tested by comparing the harmonic mean of the Bayes factor (BF) from runs under a dependent (correlation allowed) character model to that under an independent (correlation fixed to 0) model. A log(BF) >10 was considered to give very strong support for the best model. To incorporate phylogenetic uncertainty, the analysis was simulated on 50 trees sampled from the posterior distribution of trees from the phylogenetic analysis. MCMC chains were run for 10 million generations, sampling chains every 1000 generations.

## Acknowledgements

We thank Waltraud Röseler, Aziza Aust, and Hanh Witte for helping with DNA preparation, Matthias Herrmann for help with nematode collecting, and Robin Giblin-Davis for providing several nematode strains.

## Additional information

### Funding

| Funder | Author |
| --- | --- |
| Max-Planck-Gesellschaft (Max Planck Society) | Ralf J Sommer |

The funder had no role in study design, data collection and interpretation, or the decision to submit the work for publication.

### Author contributions

VS, EJR, Conception and design, Acquisition of data, Analysis and interpretation of data, Drafting or revising the article; NK, Sample collection, Acquisition of data; RJS, Sample collection, Drafting or revising the article

## Additional files

### Supplementary file

• Supplementary file 1. Two tree files and a multiple sequence alignment file.

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
