## [Decision Letter]

Thank you for sending your work entitled “Rapid diversification associated with a macroevolutionary pulse of developmental plasticity” for consideration at *eLife*. Your article has been favorably evaluated by Ian Baldwin (Senior editor), a Reviewing editor, and three reviewers.

The following individuals responsible for the peer review of your submission have agreed to reveal their identity: Merijn Kant (Reviewing editor), Fred Nijhout (peer reviewer 1), and Paul Brakefield (peer reviewer 2). A third reviewer remains anonymous.

The Reviewing editor and the reviewers discussed their comments before we reached this decision, and the Reviewing editor has assembled the following comments to help you prepare a revised submission.

The manuscript by Susoy and colleagues makes key advances in describing how the evolution of developmental phenotypic plasticity can play a major role in shaping and facilitating patterns of evolutionary diversification. It uses surveys of patterns of morphological variation relating to mouthpart and feeding function in numerous species of nematodes in combination with a more extensive and complete phylogeny to test whether the evolution of phenotypic plasticity is associated with the tempo of evolution. The value of this work is that current literature on this topic is short of empirical and quantitative studies; the submitted manuscript could thus be an important contribution.

While the data are fantastic, we were not fully convinced by the data interpretation and how the uncertainties associated with these interpretations are currently being discussed. We suggest you to textually nuance your claims by elaborating in more detail on why loss of plasticity should facilitate evolutionary diversification and how plasticity can shape evolutionary trajectories. Although this will remain largely speculative in nature, it will especially be useful to readers searching for the wider context.

Firstly, we feel that the possibility of having the plasticity/diversification order reversed is not discussed explicitly enough and the claim will gain in strength when defended more explicitly against the alternative 'quantitative genetic explanation'. We sketched a scenario in which diversity increases after plasticity disappears: if plasticity is an evolved adaptation, then it must be stabilized against genetic variation. Increased plasticity leads to reduced heritability and maybe even genetic variance, which in turn hampers response to selection. Assuming plasticity reduces the correlation between genotype and phenotype, genotypic drift could occur without affecting the phenotype. Strong environmental determination of the phenotype may also facilitate the accumulation of hidden genetic variation, i.e. cryptic genetic variation can accumulate during periods of plasticity, and could be released when the stabilizing mechanisms that produced the plasticity become (for whatever reason) inactive. In their absence the population might have drifted away from the fitness optimum or the fitness optimum might have moved away from the population (e.g., changing environment or ecological interactions), leading to increased selection pressure and thus “rapid” evolutionary change and an increase in phenotypic diversity. We would like to encourage you to dedicate a paragraph near the end of the Discussion to this and to indicate to which extent one can differentiate between different scenarios.

Secondly, we feel that the Discussion is focused too strongly on the discrete nature of the variation across the whole phylogeny while there clearly is some uncertainly about the wider relevance of your lab environments (in which you used extremes of prey items) and, consequently, the ability of this study to uncover more quantitative variability in norms of reactions involving a wider range of environments (potentially more relevant to those experienced by some of the taxa). Also, this could be discussed in an additional paragraph.

---

## [Author Response]

*While the data are fantastic, we were not fully convinced by the data interpretation and how the uncertainties associated with these interpretations are currently being discussed. We suggest you textually nuance your claims by elaborating in more detail on why loss of plasticity should facilitate evolutionary diversification and how plasticity can shape evolutionary trajectories. Although this will remain largely speculative in nature, it will especially be useful to readers searching for the wider context*.

*Firstly, we feel that the possibility of having the plasticity/diversification order reversed is not discussed explicitly enough and the claim will gain in strength when defended more explicitly against the alternative 'quantitative genetic explanation'. We sketched a scenario in which diversity increases after plasticity disappears: if plasticity is an evolved adaptation, then it must be stabilized against genetic variation. Increased plasticity leads to reduced heritability and maybe even genetic variance, which in turn hampers response to selection. Assuming plasticity reduces the correlation between genotype and phenotype, genotypic drift could occur without affecting the phenotype. Strong environmental determination of the phenotype may also facilitate the accumulation of hidden genetic variation, i.e. cryptic genetic variation can accumulate during periods of plasticity, and could be released when the stabilizing mechanisms that produced the plasticity become (for whatever reason) inactive. In their absence the population might have drifted away from the fitness optimum or the fitness optimum might have moved away from the population (e.g., changing environment or ecological interactions), leading to increased selection pressure and thus ”rapid“ evolutionary change and an increase in phenotypic diversity. We would like to encourage you to dedicate a paragraph near the end of the Discussion to this and to indicate to which extent one can differentiate between different scenarios*.

We fully agree with this suggestion and have incorporated a new paragraph (second to last of the main text), which includes a new discussion along these lines. In addition, we changed the last paragraph of the Discussion to adjust the wording and interpretation of our data.

*Secondly, we feel that the Discussion is focused too strongly on the discrete nature of the variation across the whole phylogeny while there clearly is some uncertainly about the wider relevance of your lab environments (in which you used extremes of prey items) and, consequently, the ability of this study to uncover more quantitative variability in norms of reactions involving a wider range of environments (potentially more relevant to those experienced by some of the taxa). Also, this could be discussed in an additional paragraph*.

We agree that the possibility for quantitative variability was not fully addressed and have consequently provided a more thorough explanation of this in our revised version. However, we felt that this would best be done in our elaborate Materials and methods section, where we have now included more detailed information as requested. We feel that this new information will be very helpful to the readers.